# Evaluating Machine Learning Methods for Predicting Diabetes among Female Patients in Bangladesh

**Badiuzzaman Pranto** [ID] **, Sk. Maliha Mehnaz** [ID] **, Esha Bintee Mahid** [ID] **, Imran Mahmud Sadman** [ID] **, Ahsanur Rahman** [ID] **and Sifat Momen \*** [ID]

Department of Electrical and Computer Engineering, North South University, Dhaka 1229, Bangladesh;
badiuzzaman.pranto@northsouth.edu (B.P.); maliha.mehnaz@northsouth.edu (S.M.M.);
esha.mahid@northsouth.edu (E.B.M.); mahmud.sadman@northsouth.edu (I.M.S.);
ahsanur.rahman@northsouth.edu (A.R.)
* Correspondence: sifat.momen@northsouth.edu

**Abstract:** Machine Learning has a significant impact on different aspects of science and technology including that of medical researches and life sciences. Diabetes Mellitus, more commonly known as diabetes, is a chronic disease that involves abnormally high levels of glucose sugar in blood cells and the usage of insulin in the human body. This article has focused on analyzing diabetes patients as well as detection of diabetes using different Machine Learning techniques to build up a model with a few dependencies based on the PIMA dataset. The model has been tested on an unseen portion of PIMA and also on the dataset collected from Kurmitola General Hospital, Dhaka, Bangladesh. The research is conducted to demonstrate the performance of several classifiers trained on a particular country's diabetes dataset and tested on patients from a different country. We have evaluated decision tree, K-nearest neighbor, random forest, and Naïve Bayes in this research and the results show that both random forest and Naïve Bayes classifier performed well on both datasets.

**Keywords:** diabetes prediction; PIMA dataset; kurmitola general hospital; machine learning; classification

---

## 1. Introduction

A major public health problem, diabetes, is the surfeit rise of sugar level in blood and occurs when pancreas does not produce insulin, or in spite of the pancreas producing insulin, the body cannot use it effectively [1]. It is the root cause of many associated health diseases. Diabetic peripheral neuropathy [2], for example, is a form of nerve pain caused by diabetes. Risk of eye problems increases due to the presence of diabetes and can lead to diabetic retinopathy [3] and diabetic macular edema [3]. Diabetic nephropathy [4] which is the primary cause behind kidney failure is a consequence of diabetes. People with diabetes are more prone to coronary heart diseases and have heart attacks [5]. Polycystic Ovary Syndrome (PCOS) [6], a physical condition hampering the overall ovulation process, also increases the chances of having diabetes. There are three main categories of diabetes:

- **Type 1 diabetes** [7] occurs mostly to children and adolescents. In this case, the body produces very little or no insulin at all. As a result, daily insulin injections are needed to keep glucose levels under control. Frequent urination, sudden weight loss, abnormal thirst, constant hunger, blurred vision, and tiredness are common symptoms of this kind of diabetes. This can be treated with the help of insulin therapy.

- **Type 2 diabetes** [8] is more prevalent in adults (90% cases). The body does not fully respond to insulin resulting in higher glucose levels. Obesity, unhealthy diet, high blood pressure, and physical inactivity are considered to be major risk factors that lead to type 2 diabetes.

Insulin injections are required when oral medication is not sufficient enough to control blood sugar levels.

- **Gestational Diabetes Mellitus (GDM)**, or simply gestational diabetes consists of high blood pressure during pregnancy and can cause health complications to both mother and children. It usually disappears during the pregnancy stage but the affected ones along with their children have a risk of developing Type 2 diabetes in their later life. According to a survey in 2017 [9], approximately 204 million women suffers from GDM. About 21.3 million live births had some form of hyperglycemia in pregnancy, among which about 85.1% occurred due to gestational diabetes. GDM typically affects around one out of seven births.

Diabetes, of all types, can result in different complications in the body and also increase the overall risk of premature death. A recent research of 2017 shows that people with PCOS have an increased chance of having Type 2 Diabetes at a later age [10]. According to the International Diabetes Federation (IDF) atlas 2019, one out of 11 adults (20–79 years) have diabetes which can be approximated as 463 million people [11]. The global health report on diabetes from WHO shows that in addition to the 1.5 million deaths from diabetes in 2012, another 2.2 million deaths are resulted from cardiovascular and other diseases due to increased blood sugar levels. For the past three decades, diabetes has been increasing steadily and is growing more rapidly in low and middle-income countries [12].

The WHO report, in 2016, shows that 1.6 million deaths occurred directly due to diabetes [13] and in 2012, high blood glucose resulted in 2.2 million deaths.

## 1.1. Diabetes in Bangladesh

Bangladesh is a densely populated developing country located in South Asia, covering an area of 147,750 square km. With an estimation of 165.2 million people in 2019, Bangladesh is considered to be the 8th most populous country in the world [14]. Diabetes has unquestionably been a serious threat in a developing country like Bangladesh, where a huge portion of the population is oblivious to the detrimental effect of diabetes [15]. There has been a rapid increase in diabetes rates in Bangladesh owing to the remarkable socioeconomic, demographic, and epidemiological changes over the past few decades [16]. Around 80 lac people in Bangladesh already suffer from this global health problem [17]. In 2016, Bangabandhu Sheikh Mujib Medical University(BSMMU) conducted a survey on 2000 adults in Dhaka slums and found out that 19% of the adults, among whom 15.6% male and 22.5% female, had already suffered from diabetes [17]. Around 45% of people from the rural area take medical advice and assessments from unqualified doctors, assistants, and midwives. These patients are at a risk of moderate to severe complications if they are already suffering from diabetes and not taking necessary medications [18]. Availability of medicine, basic technologies, and necessary procedures for monitoring diabetes in primary health care has been found to be very limited in Bangladesh [19]. According to the WHO Diabetes country profile, in 2016, the number of deaths among males, due to diabetic conditions, aged from 30 to 69 was 6060, and 20,000 people died due to high blood glucose. Furthermore, 4760 females died from diabetes and 10,600 suffered from deaths related to high blood glucose levels (all aged between 30 to 69) [19]. Diabetes [20] is a long term disease that cannot be cured. However, the symptoms and later complications can be controlled by necessary treatments and a healthy lifestyle. Thus, this research has substantial importance as it enables detecting diabetes at early stage.

## 1.2. Applicability of Machine Learning in Detecting Diabetes

Machine Learning (ML) has now become increasingly popular and has been reported as one of the most effective methods in a wide range of applications in preventive healthcare [21]. It has associated advantages such as relatively low-cost computation, robustness, generalization ability and high performance [22]. With the development of medical devices, equipment and tools, advanced knowledge can be gained in the disease diagnosis field. Computer-assisted decision making, i.e., Machine Learning aids humans by processing complex medical datasets and analyzing them

to provide clinical insights [23,24]. The knowledge extraction from data is a crucial factor for the prediction and diagnosis of disease in the medical industry [25,26]. Through acquisition of required data and then necessary training and testing, some observations are obtained that can help reach a conclusion. In this article, we aim to detect whether any female patient residing in Bangladesh is suffering from diabetes or not. Inductive learning best suits this kind of work. In inductive learning, a learner learns some rules from observation of a set of instances. In inductive learning, the output can be predicted for new samples in the future through generalization and mapping. Machine Learning is one of the most conducive approaches to be applied for this.

*1.3. Research Goal*

In this article, we aim to show the analytical results of how different physical factors and conditions can affect as well as give rise to the chances of diabetes considering the female population. In this article, the analytical results of how different physical factors and conditions can contribute and give rise to the chances of diabetes considering the female population have been interpreted. Furthermore, we wanted to assess how classifiers built on the PIMA dataset perform on both Indian and Bangladeshi diabetic patients. Our research objective is to answer the following research question:

Can Machine Learning techniques be applied to predict the occurrence of diabetes among female patients in Bangladesh? Can Machine Learning techniques be applied to predict diabetes in patients of Bangladesh based on the PIMA Indian dataset?

Data availability is a huge concern, predominantly noticed in developing countries like Bangladesh [27]. Accomplishment of the above-mentioned points can conclude that a single dataset taken from India can be used to train Machine Learning models which can then be applied to the female population of Bangladesh in diagnosis and detection of diabetes. Thus the problem of data unavailability can be resolved to some extent.

This will give the mass population an in-depth knowledge and a close overview about the dependencies of various health conditions so that they can be aware and take necessary precautions in order to avoid the chances of diabetes occurring at an early age.

The rest of the article is organized as follows: In Section 2, we review related work carried out by researchers. This is followed by the analysis and description of the dataset in Section 3 and then the methodology adopted in this article described in Section 4. Section 5 contains the results of the model as well as the discussions followed by the concluding remark in Section 6.

## 2. Related Work

The last decade has witnessed the use of various Machine Learning techniques by the researchers to predict diabetes. Several classifiers including random forest, instance-based classifier (IBK), J48, as well as fuzzy approaches have been used to detect diabetes. Some of the papers reviewed below have used the dataset available at the UCI Machine Learning repository [28] based on India known as the PIMA dataset [29]. We have focused on these papers since our work is also done depending on this dataset. In most research papers it is observed that in 768 instances, a total of nine attributes were used as input variables and one output variable was considered for prediction of diabetes according to the dataset. Besides that, reviews on papers based on iridology, on protein–protein interactions, and also on a dataset collected from Noakhali, Bangladesh are provided.

*2.1. Research on PIMA Dataset*

In one research, the authors [30] investigated different types of Machine Learning classification algorithms and made a comparative analysis. Logistic regression was found to yield the highest accuracy of 78.01%. Based on their results, logistic regression, along with sequential minimal optimization (SMO) and multilayer perceptron (MLP), could be used to predict the onset of diabetes with an accuracy of 78%. Bagging, Naïve Bayes, and Support Vector Machine (SVM) also performed well.

Ravi Sanakal [31] used the fuzzy c means (FCM) clustering and support vector machine to get prediction for diabetes mellitus. The PIMA dataset was used in the investigation. The highest accuracy (94.3%) was obtained using FCM. On the contrary, SVM resulted in an accuracy of 59.5% which was quite low.

A comparative analysis has been done in one [32] research to predict diabetes with a high accuracy outcome using multiple Machine Learning techniques. The authors have precisely evaluated the performance of Naïve Bayes, C5.0 decision tree, logistic regression, and Support Vector Machine (SVM) using the PIMA dataset. Depending on the accuracy results, the random forest classifier gave an accuracy of about 74.67%, which was the highest among the other classifiers used for the research.

Gujral and colleagues [33] used support vector machines (SVM) and decision tree to detect as well as predict the risk of diabetes. Detection was done successfully with the SVM classifier with an accuracy of 82%.

Radja and Emanuel conducted a series of systematic experiments to evaluate the performance of supervised Machine Learning algorithms using different data sizes for diabetes prediction [34]. Their purpose was to determine which types of algorithms were most accurate in predicting diabetes using small amount of data, and which algorithms were the most accurate in measuring large amounts of data. After evaluating several classifiers, they concluded that the best algorithm that can be used to help decide to diagnose a disease is the SVM algorithm with an accuracy value of 77.3%.

In another work [35], the PIMA dataset was trained to give the predetermined output using different Machine Learning approaches. SVM was used to construct a hyper-plane that divided the entire dataset into several categories. They used principal component analysis (PCA) in order to find out the maximum co-relation in the dataset in order to select the feature. Furthermore, they evaluated the decision tree classifier and then finally made a conclusion that SVM provides the highest accuracy of 82%.

Kadhm and colleagues carried out K-means clustering on the PIMA dataset and divided it into 10 clusters to get the most accurate result [36]. Each cluster had a number of healthy and diabetic samples. Their proposed work gave an accuracy of around 98.7% which was significantly higher than the existing used algorithms.

### 2.2. Research Based on Image Analysis

Machine Learning techniques have also been applied for the diagnosis of diabetes on the basis of iridology. The authors [37] used iris images of people in order to identify any form of changes in their iris and thus differentiate between the diabetic and non-diabetic patients using various Machine Learning techniques. Feature extraction was done using the gray-level co-occurrence matrix (GLCM) and then Naïve Bayes, K-nearest neighbor, ensemble learning, SVM, and random forest have been applied. For the image pre-processing part, image enhancement, localization and normalization were performed. A maximum accuracy of 85.6% along with the specificity of 0.900 and sensitivity of 0.800 was obtained.

### 2.3. Research Based on PPI Prediction

Researchers, in the past, have also conducted experiments to predict protein–protein interactions in type-2 diabetes. The authors [38] used feedforward neural network (FNN) to predict protein-protein interactions (PPIs) of type-2 diabetes in this research. They observed the impact of different activation functions, the number of units per hidden layers, and number of hidden layers themselves to estimation error. With the configuration of the rectifier activation function, seven hidden layers, and 36 units per hidden layers , the model predicted a PPI network with a predicted combined score of 92.2%.

### 2.4. Research Based on Dataset from Bangladesh

We have come across one article [39], where Machine Learning has been applied to predict the severity of diabetes from the perspective of a district in Bangladesh and determine the

significant features of it. The entire dataset was collected from Noakhali Diabetes Association, Bangladesh, where it had around 220 instances with 13 attributes. This article solely used the dataset from Bangladesh to build a prediction model using four decision tree based classifiers, which were CDT, J48, NBtree, and REPTree. They analyzed predicted outcomes and evaluated the best classifier using individual evaluation metrics such as kappa statistics, recall, precision, f-measure, AUROC, and RMSE. CDT (unpruned) classifier could come up with a highest accuracy of 96.98%.

There have been very few works where the training dataset taken from one country has been tested on a dataset collected from another country. As observed, there are many researchers who worked on the area pertaining to Machine Learning and diabetes and many are still working on it. Some came up with remarkable results as well. A key difference in our research that distinguishes it from previous work lies in finding classifiers that are best suited for female diabetic patients in Bangladesh. Such work, to the best of our knowledge, has not been conducted before.

## 3. Data

### 3.1. Data Acquisition and Description

The PIMA Indians Diabetes dataset [29], obtained from Kaggle , has been originally collected from the National Institute of Diabetes, Digestive and Kidney Diseases, India. Alongside this, we collected more real data from the general ward of Kurmitola General Hospital, Bangladesh (KGH) in order to test the ML model and predict diabetes for Bangladeshi patients. We approached a team of intern doctors who were involved in the data collection process. They conducted a short discussion with the patients and after proper consent from them, the doctors agreed to provide us the relevant data. In all, it took us around three weeks in the month of November 2019 to collect all the relevant data. Moreover, collecting data from Kurmitola General Hospital matching all the features with the PIMA dataset turned out to be an enormous challenge for us. Firstly, we faced a few hindrances while looking for similar attributes to the PIMA dataset. Somehow, we could manage to have a few closely related attributes. We had to do conversions of the instances to place them in the same unit. In addition, the data had to be collected manually and later converted to digital file format, i.e., in the csv format.

Table 1 shows the summary of both datasets. It is evident from Table 1 that there exists a class imbalance due to the existence of a higher number of negative values than positive ones. To address this problem, precision, recall, F1 score, and area under ROC curve (AUC) score has been considered along with accuracy rather than just accuracy itself.

Both datasets consist of female patients which is why the number of times of pregnancy is one of the most important features for our analysis. We have evaluated several Machine Learning classifiers on these particular datasets to predict whether a female patient is suffering from diabetes or not. Tables 2 and 3 describes the attributes of PIMA dataset and Kurmitola dataset respectively.

**Table 1.** Dataset summary.

| Dataset | Number of instances | Number of features | Positive | Negative |
|---|---|---|---|---|
| PIMA | 768 | 8 | 268 | 500 |
| Kurmitola Hospital | 181 | 4 | 50 | 131 |

**Table 2.** Pima dataset features.

| SL | Feature Name | Description | Min val | Max val | Mean |
|----|--------------|-------------|---------|---------|------|
| 1 | Number of pregnancy | Number of times pregnant | 0 | 17 | 3.85 |
| 2 | Glucose concentration | 2-h oral glucose test (mg/dL) | 0 | 199 | 120.89 |
| 3 | Blood Pressure | Diastolic blood pressure (mm Hg) | 0 | 122 | 69.11 |
| 4 | Skin thickness | Triceps skin fold thickness (mm) | 0 | 99 | 20.54 |
| 5 | Serum Insulin | 2-H serum insulin (mu U/mL) | 0 | 846 | 79.80 |
| 6 | BMI | Body mass index (kg/m$^2$) | 0 | 67.10 | 31.99 |
| 7 | Diabetes Pedigree Function | Diabetes in family history | 0.08 | 2.42 | 0.47 |
| 8 | Age | Age in Years | 21 | 81 | 33.42 |

**Table 3.** Kurmitola General Hospital (KGH) dataset features.

| SL | Feature Name | Description | Min val | Max val | Mean |
|----|--------------|-------------|---------|---------|------|
| 1 | Number of pregnancy | Number of times pregnant | 1 | 5 | 1.97 |
| 2 | Glucose concentration | 2-h oral glucose test (mg/dL) | 82.80 | 160.20 | 113.79 |
| 3 | BMI | Body mass index (kg/m$^2$) | 19.10 | 37.20 | 28.46 |
| 4 | Age | Age in Years | 17 | 58 | 26.76 |

*3.2. Data Analysis and Feature Selection*

Figure 1, known as the correlation matrix, describes how the features are correlated to each other. It also illustrates which feature is contributing most to the outcome (predictions). According to the matrix, the outcome is strongly dependent on the glucose level. If we recall the medical definition of diabetes [1], "diabetes occurs because of the extra glucose or sugar level in our blood." Figure 1 exactly visualizes and demonstrates the same interpretation of the medical definition. The glucose level has a strong positive correlation with the outcome which is why the probability of being exposed as diabetic patient increases with the increment of glucose level in the blood cell.

As we previously mentioned in Section 1, "The blood sugar level increases because insulin is not properly produced or the produced insulin is not properly used by our body" [1]. When the body does not convert enough glucose, blood sugar levels remain high. Insulin helps the cells to absorb glucose, reducing blood sugar, and providing the cells with glucose for energy [1]. This further anticipates a correlation between glucose and insulin. In this case, Figure 1 again proves the validity of the dataset by visualizing the correlation between the features as expected. Pregnancy and age are also positively correlated to each other since the number of pregnancies generally increases with age.

This research is conducted to find a Machine Learning model based on the PIMA dataset and then evaluate it on Kurmitola Hospital dataset. To do that, we needed to match the features of both datasets. Unfortunately, while collecting data from Kurmitola Hospital, it was not possible to collect all the features similar to PIMA dataset. Based on the combined decision of the common parameters available between PIMA dataset and KGH dataset and the correlation matrix, we finally proceeded with four features which are as follows

- Number of times the patient got pregnant,
- Blood sugar or glucose,
- Body mass index (BMI) and
- Age of the patient.

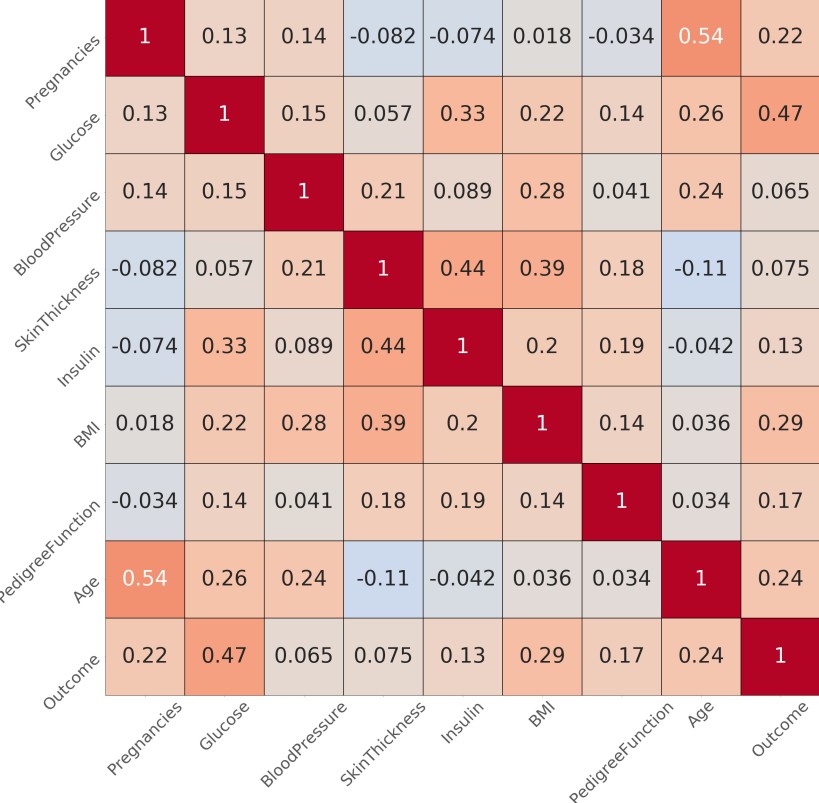

**Figure 1.** Correlation matrix of PIMA Dataset.

A box and whisker plot is a simple way of summarizing statistical data. It provides a five-number summary of the data available namely, the minimum, first quartile, median, third quartile, and maximum. From Figures 2 and 3, it is observed that for varying features, different box-plots have been derived for the positive and negative outcomes of PIMA as well as Kurmitola hospital dataset. The summary of both figures are shown in Tables 4 and 5. Since the PIMA dataset will be used in training phase, the presence of *zero* values in glucose (blood sugar level) and BMI needs to be cleaned during the data pre-processing phase.

When comparing the two datasets, it is evident that there exist a lot of similarities between them. Both PIMA and Kurmitola Hospital datasets are distributed over similar ranges. For both the datasets, attributes are mostly found to be normally distributed. This is because for most attributes, the mean and median values are very close. Pregnancies, in the PIMA dataset, is observed to be slightly positively skewed and that in the Kurmitola dataset, slightly negatively skewed. Another key difference that we have noticed is that the mean values for the attributes in Kurmitola Hospital generally tends be a little lower than that of the corresponding PIMA dataset.

**Table 4.** Box-plot summary of PIMA dataset.

| SL | Box-plot Attribute | Outcome | Min. | 1st Quartile | Median | 3rd Quartile | Max. |
|---|---|---|---|---|---|---|---|
| 1 | Sugar level | Positive | 0 | 119.0 | 140.0 | 167.0 | 199.0 |
| 2 | Sugar level | Negative | 0 | 93.0 | 107.0 | 125.0 | 197.0 |
| 3 | No. of pregnancies | Positive | 0 | 1.75 | 4.0 | 8.0 | 17.0 |
| 4 | No. of pregnancies | Negative | 0 | 1.0 | 2.0 | 5.0 | 13.0 |
| 5 | BMI | Positive | 0 | 30.8 | 34.25 | 38.78 | 67.1 |
| 6 | BMI | Negative | 0 | 25.4 | 30.05 | 35.3 | 57.3 |
| 7 | Age | Positive | 21.0 | 28.0 | 36.0 | 44.0 | 70.0 |
| 8 | Age | Negative | 21.0 | 23.0 | 27.0 | 37.0 | 81.0 |

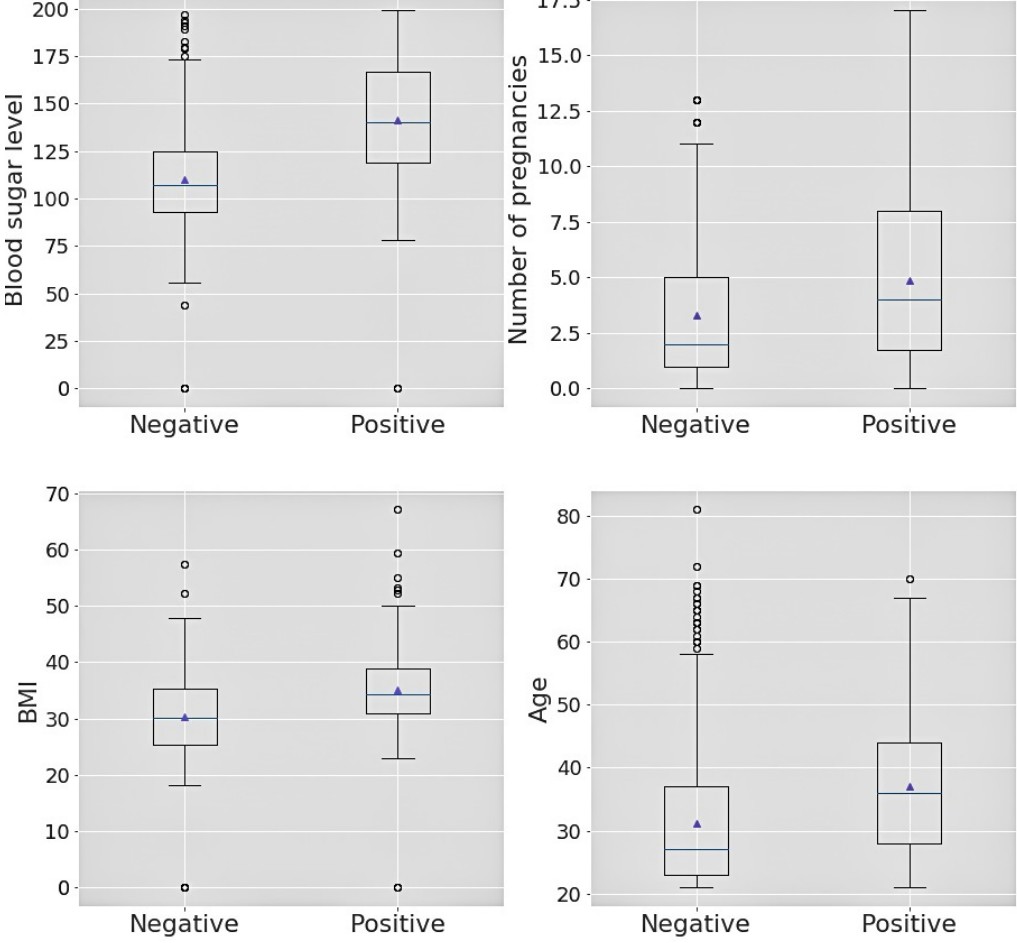

**Figure 2.** Box-plot visualization of diabetic and non-diabetic people in PIMA dataset.

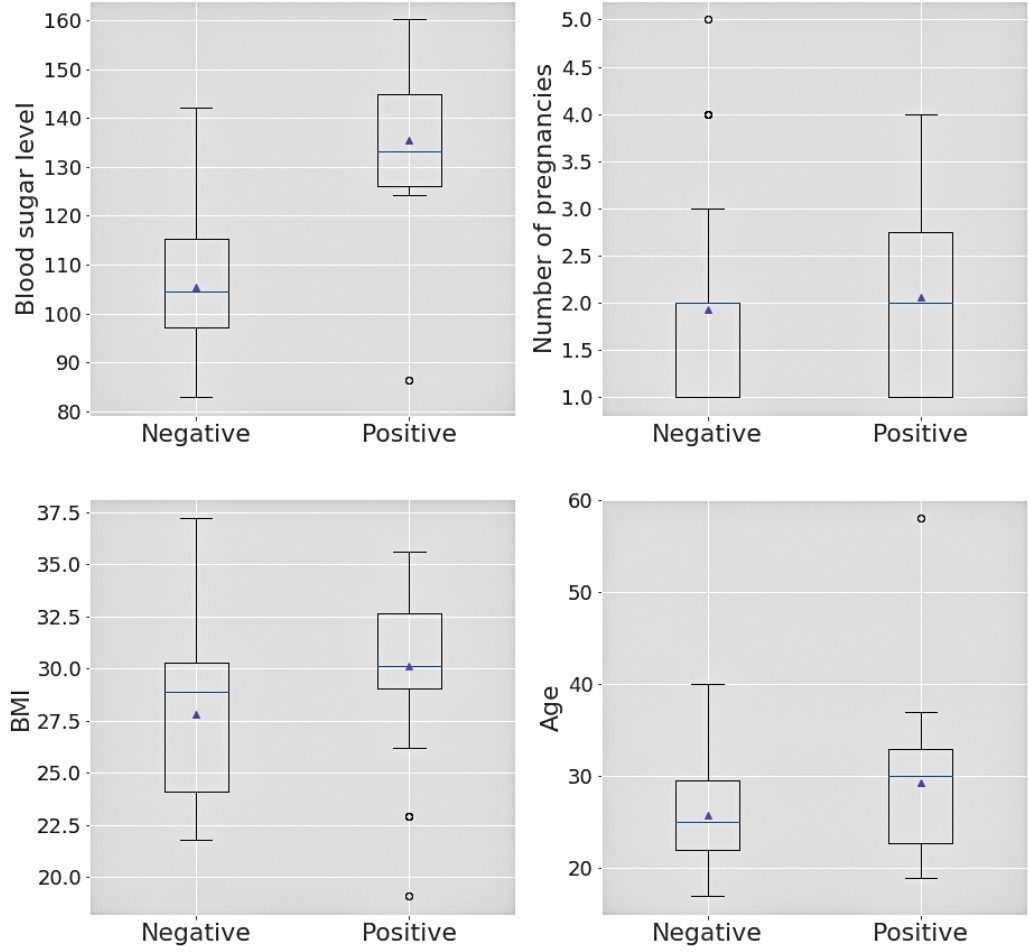

**Figure 3.** Box-plot visualization of diabetic and non-diabetic people in Kurmitola Hospital dataset.

**Table 5.** Box-plot summary of Kurmitola hospital dataset.

| SL | Box-plot Attribute | Outcome | Min. | 1st Quartile | Median | 3rd Quartile | Max. |
|---|---|---|---|---|---|---|---|
| 1 | Sugar level | Positive | 86.4 | 126.0 | 133.2 | 144.9 | 160.2 |
| 2 | Sugar level | Negative | 82.8 | 97.2 | 104.4 | 115.2 | 142.2 |
| 3 | No. of pregnancies | Positive | 1 | 1 | 2 | 2.75 | 4 |
| 4 | No. of pregnancies | Negative | 1 | 1 | 2 | 2.75 | 5 |
| 5 | BMI | Positive | 19.1 | 29.0 | 30.1 | 32.7 | 35.6 |
| 6 | BMI | Negative | 21.8 | 24.1 | 28.9 | 30.3 | 37.2 |
| 7 | Age | Positive | 19 | 22.75 | 30.0 | 33.0 | 58 |
| 8 | Age | Negative | 17 | 22.0 | 25.0 | 29.5 | 40 |

## 4. Methodology

The methodology, as outlined in Figure 4, was followed while evaluating the Machine Learning models. First, we took the PIMA dataset and then pre-processed the dataset. After pre-processing, the dataset was split into train and test sets. ML algorithms were applied on the training set to create ML models. The performance of the models were then evaluated using the test set from the PIMA dataset as well as that of the Kurmitola dataset.

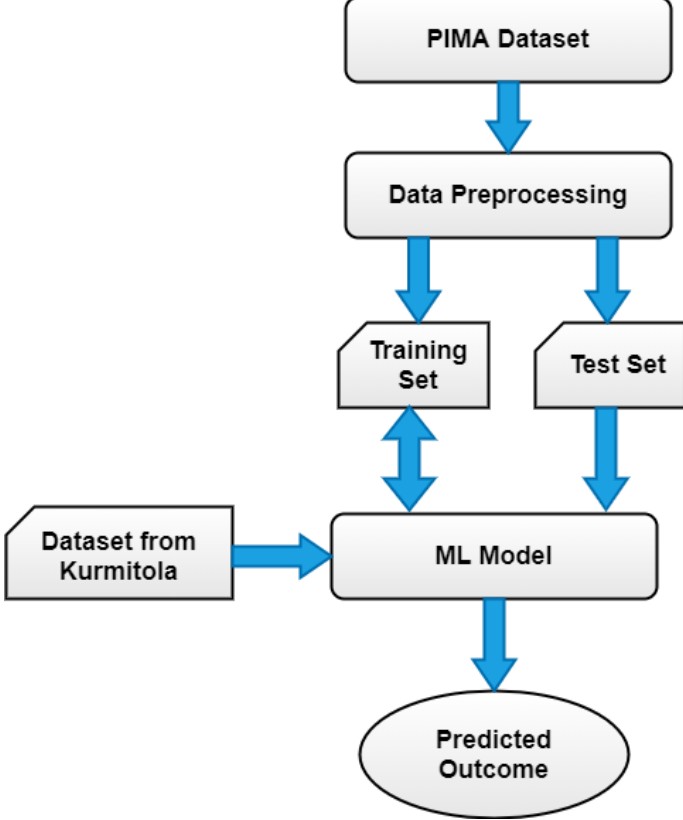

**Figure 4.** Flowchart for predicting diabetes using Machine Learning.

### 4.1. Data Pre-Processing

First of all, we analyzed the data for errors and found some issues in general. Data in the real world are mostly inconsistent or noisy and require to be pre-processed either by filling the missing values or by dropping them. As we have witnessed before in Figure 2, there were zero values in blood glucose level and BMI which is not possible. We have filled these values with the mean of their corresponding feature. Kurmitola General Hospital's dataset had the feature glucose in mmol/L which has been converted into mg/dL to match with the unit from PIMA. Since we couldn't collect all eight features for Kurmitola Hospital which are available on PIMA so we ultimately continued with four features commonly available in both datasets.

The PIMA dataset was divided on a ratio of 70:30 for creating training vs test split because we found such split ratio common in the literature [40]. A three fold cross validation technique was applied on the test set section. Cross validation is a statistical resampling procedure that helps to refine model so that prediction can be made with higher confidence. This is typically suited when the dataset used is small. Features of both datasets contain large values or had a massive difference compared to other feature-values which could result in bias towards the features containing higher values. In order to tackle this issue, we normalized all features using Equation (1). This resulted in values lying between 0 and 1.

$$X_{\text{norm}} = \frac{X - X_{\text{min}}}{(X_{\text{max}} - X_{\text{min}})} \tag{1}$$

### 4.2. Machine Learning Model Implementation

In this research, we have used decision tree, K-nearest neighbor, random forest, and Naïve Bayes to train four different classifiers, and then evaluated the classifiers to find the performance on Bangladeshi data. These classifiers have been explained shortly in Sections 4.3.1–4.3.4. Implementation of the

classifiers, calculation, and results generation have been conducted using Sklearn [41], a Python-based Machine Learning library while figure and graph generation have been done using Matplotlib [42] and Seaborn library [43].

After completing the pre-processing stage, the training set consisting of 538 instances was used to train the classifiers. Three-fold cross-validation was performed on this training set to find suitable values of hyperparameters. The size of the dataset was small and this governed our choice of 3-fold cross-validation. In each cross-validation test, the classifiers were trained with 358 instances and then tested with 179 instances. Once the hyperparameters were tuned, we fed data to our model for the mentioned classifiers with the best hyperparameters. Hyperparameter tuning for each classifier has also been explained in Sections 4.3.1–4.3.3.

### 4.3. Algorithms and Their Hyperparameter Tuning

In this section, we describe the algorithms that have been used in our research. Some algorithms require hyperparameter tuning before training the classifiers. For example, we need to choose a value of *K* for K-NN, maximum depth of a tree for decision tree and numbers of trees for random forest. The choice of the classifier used in our work is strongly inspired by the fact that these classifiers have been effective in preventive healthcare. Algorithm 1 shows how we selected the best hyperparameter for each of these classifiers. The following sections describe different Machine Learning algorithms and their corresponding hyperparameters.

---

**Algorithm 1:** Algorithm for hyperparameter tuning

---

   1. $n \leftarrow 0$;
   2. *hyperparameter* $\leftarrow 1$;
   3. **while** $n \leq 1000$ **do**
       4. Perform a 3 fold cross-validation on PIMA train set;
       5. Record the average accuracy for 3 fold cross-validation;
       6. Increment hyperparameter by 1;
       7. Increment n by 1;
   **end**
   8. Choose the final hyperparameter which gives the highest average cross-validation accuracy;

---

#### 4.3.1. Decision Tree

Decision tree learning employs techniques that are typically used to approximate a discrete-valued target function. The learned function can be represented as sets of if-else/then rules. This aids us to understand how a decision tree classifier reaches a decision. Instances in decision trees are classified by sorting them down the tree starting from the root node and ending to some leaf node. Each node specifies a test of some feature of the instance, and each branch descending from that node corresponds to one of the possible values for this feature. The trees select the root node based on a statistical calculation called information gain.

The information gain of a node can be measured through the Gini index or entropy [40]. We have used entropy to calculate the information gain. Entropy tells us how impure a collection of data is. The term impure here defines non-homogeneity. In other words, we can say that entropy is the measurement of homogeneity. It returns us the information about an arbitrary dataset that how impure/non-homogeneous the dataset is [44]. The decision tree implemented in this research work is based on CART(classification and regression trees algorithm) [45].

$$Entropy(S) = -(P_{\oplus}log_2P_{\oplus} + P_{\ominus}log_2P_{\ominus}) \tag{2}$$

$$Gain(S, A) = Entropy(S) - \sum_{v \in Values(A)} \frac{|S_v|}{|S|} Entropy(S_v) \tag{3}$$

Maximum depth (length of the longest path from the root of a tree to its leaf) was considered as the hyperparameter. The tree overfitted on the training set while increasing the depth of the tree. If we set the maximum depth too high then it increased the chances of overfitting the training data without capturing useful patterns as well as resulting in testing error. This scenario is visualized in Figure 5. In order to fix these issues, we observed the average of 3-fold cross-validation accuracy and found out that the best average accuracy of 3-fold cross-validation was at a maximum depth of 2; the red line in Figure 5. We used the GridSearchCV [41] library to verify the maximum depth for best average cross-validation accuracy. GridSearchCV uses exhaustive search over specified parameter values for an estimator.

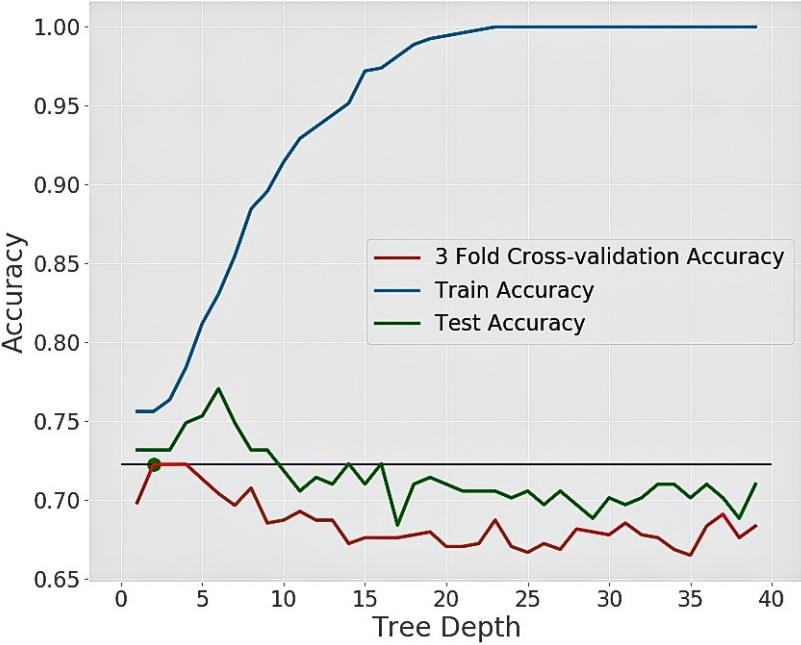

**Figure 5.** Hyperparameter tuning for decision tree.

### 4.3.2. K-Nearest Neighbor

Next, we have used the K-Nearest Neighbor classifier which is also known as instance-based classifier and belongs to a family of learning methods called lazy learning. K-NN classifier can be used for both classification and regression. However, it is best suited for classification problems. In K-nearest neighbor (KNN), the classifier evaluates distance from the test data point (also known as the query data point) to all the data points in the training set. Following this, it finds the K nearest neighbors to this test data point. A simple voting is then conducted between the K nearest data points to decide on the class that the classifier predicts. We have used Euclidean distance mentioned in Equation (4) to measure the distance.

$$d(x,y) = \sqrt{\sum_{i=1}^{n} (x_i - y_i)^2} \tag{4}$$

We need to choose a tuned value for *K* (the number of neighbors) so that our model does not overfit or underfit. For analyzing this problem, we plotted test, train, and the average accuracy of 3 fold cross-validation on Figure 6. This figure illustrates that error is Zero in training accuracy while K = 1 but the model over fits on the training set. On the other hand, while the value for *K* increases,

training accuracy reduces. The red line in Figure 6 indicates the average cross-validation accuracy of 3 fold cross-validation and it is visible that the accuracy is best at K = 16.

In order to confirm the value of K, we used the GridSearchCV library in Sklearn [41] for hyperparameter tuning. In both cases, it was found out that the value of K being 16 gave the best cross-validation accuracy.

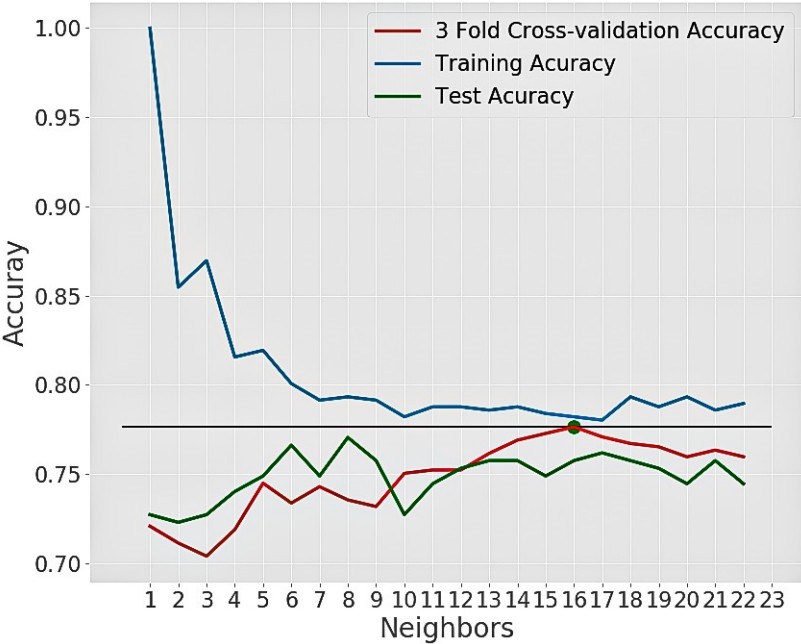

**Figure 6.** Hyperparameter Tuning for K-nearest neighbor (KNN).

### 4.3.3. Random Forest

Random forest is an ensemble of decision trees, usually trained via the bagging (or sometimes pasting) method [40]. It is one of the most efficient ensemble methods to get more precise predictions. It incorporates more diversity and reduces variances. The algorithm of a random forest is given in Algorithm 2:

---

**Algorithm 2:** Algorithm for building random forest

---

1. choose $T$: number of trees to grow.
2. choose $m$: number of features to be used to learn each tree.
3. $i \leftarrow 1$
4. **while** $i \leq T$ **do**
   5. Prepare a training subset from the given training set via random sampling with replacement.
   6. Prepare a feature subset by randomly choosing $m$ features.
   7. Learn a decision tree using those selected features and training subset.
   8. $i \leftarrow$ i+1
**end**
9. Use majority voting among all the trees to predict the outcome.

---

We chose $T = 800$ individual trees to grow and then used majority voting among all the trees. All 800 trees were based on CART algorithm [45]. The hyperparameter was decided with the same approach followed by KNN and decision tree (GridSearchCV [41] with random hyperparameter and select the one which gives the best average cross-validation accuracy).

### 4.3.4. Naïve Bayes

Naïve Bayes classifier is a classification algorithm that follows the Bayes' theorem. It paves a way so that the probability of a hypothesis can be measured provided prior knowledge. Equation (5) shows how Naïve Bayes classifier makes its decision.

$$
\begin{aligned}
P(Y|X) &= \frac{P(X|Y)P(Y)}{P(X)} \\
&= \frac{P(X_1, X_2...X_n|Y)P(Y)}{P(X)}
\end{aligned}
\tag{5}
$$

Here $Y$ is the class variable that we wanted to predict and $X$ is a dependent vector of $n$ attributes such that $X =< X_1, X_2...X_n >$. Naïve Bayes classifier assumes that a predictor on a given class is independent of the values of other predictors. This conditional independency results in 6.

$$
P(X_1, X_2, ...X_n|Y) = \prod_{i=1}^{n} P(X_i|Y)
\tag{6}
$$

The naïve bayes classifier does not require any hyperparameter tuning.

## 5. Results and Discussions

In this section, we showed the evaluation performance of classifiers and later on discussed the results. Figures 7 and 8 displays the confusion matrices for PIMA and Kurmitola Hospital dataset respectively. Many valuable information can be extracted from a confusion matrix. We calculated the accuracy, recall, precision and F1 score from these confusion matrices using Equations (7)–(10) then recorded the performance scores in Table 6.

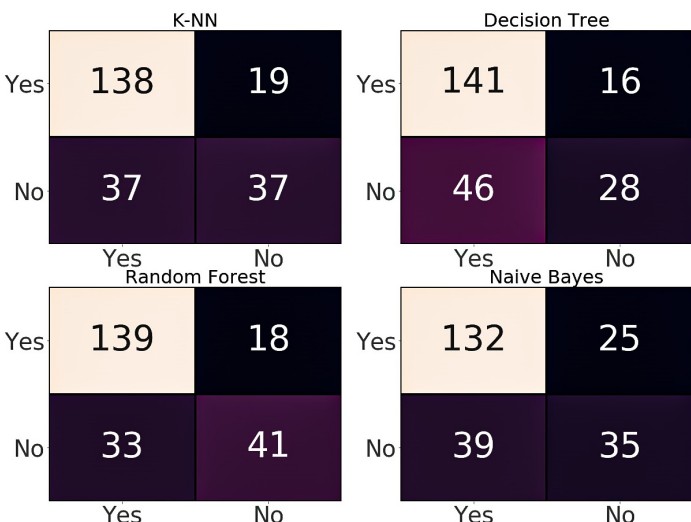

**Figure 7.** Confusion matrix comparison of four classifiers resulted from PIMA test set evaluation.

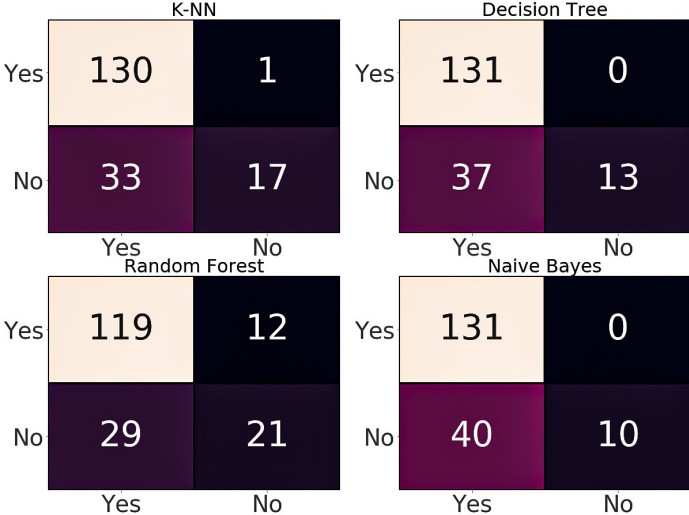

**Figure 8.** Confusion matrix comparison of four classifiers resulted from Kurmitola Hospital dataset evaluation.

$$Accuracy = \frac{TP + TN}{TP + FP + TN + FN} \tag{7}$$

$$Recall = \frac{TP}{TP + FN} \tag{8}$$

$$Precision = \frac{TP}{TP + FP} \tag{9}$$

$$F1\ score = \frac{2 * recall * precision}{recall + precision} \tag{10}$$

Here, TP = True Positive

TN = True Negative

FP = False Positive

FN = False Negative

Figures 9 and 10 show the ROC curve comparison of our four classifiers, evaluated by PIMA test set as well as the Kurmitola Hospital dataset.

ROC curve is a performance measurement for classification problem. It is plotted with True Positive Rate (TPR) against the False Positive Rate (FPR) where TPR is on $y$-axis and FPR is on the $x$-axis. The area under the ROC curve, called AUC, represents the degree of separability which provides a notion of how much a model is capable of distinguishing between classes. Higher AUC indicates better model at predicting classes. From Table 6, it is evident that the random forest performed best while testing with PIMA dataset and Naïve Bayes performed best while testing with Kurmitola Hospital dataset because in both cases, the highest area under the ROC curve were obtained.

Figure 11 displays the visualization of decision tree obtained by hyperparameter tuning from Section 4.3.1 and Table 7 exhibits the comparison of existing works with proposed work on PIMA dataset (all the existing works are trained and tested on PIMA dataset).

**Table 6.** Comparison of classification reports evaluated by PIMA test set and Kurmitola dataset.

| Dataset | Classifier Name | Accuracy | Precision | Recall | F1-score | AUC |
|---|---|---|---|---|---|---|
| PIMA test set | KNN | 0.757 | 0.79 | 0.88 | 0.83 | 0.80 |
| | Decision Tree | 0.731 | 0.75 | **0.90** | 0.82 | 0.71 |
| | Random Forest | **0.779** | **0.81** | 0.89 | **0.84** | **0.83** |
| | Naïve Bayes | 0.722 | 0.77 | 0.84 | 0.80 | 0.80 |
| Kurmitola Hospital | KNN | **0.812** | **0.80** | 0.99 | **0.88** | 0.76 |
| | Decision Tree | 0.795 | 0.78 | **1.00** | **0.88** | 0.71 |
| | Random Forest | 0.773 | **0.80** | 0.91 | 0.85 | 0.81 |
| | Naïve Bayes | 0.779 | 0.77 | **1.00** | 0.87 | **0.84** |

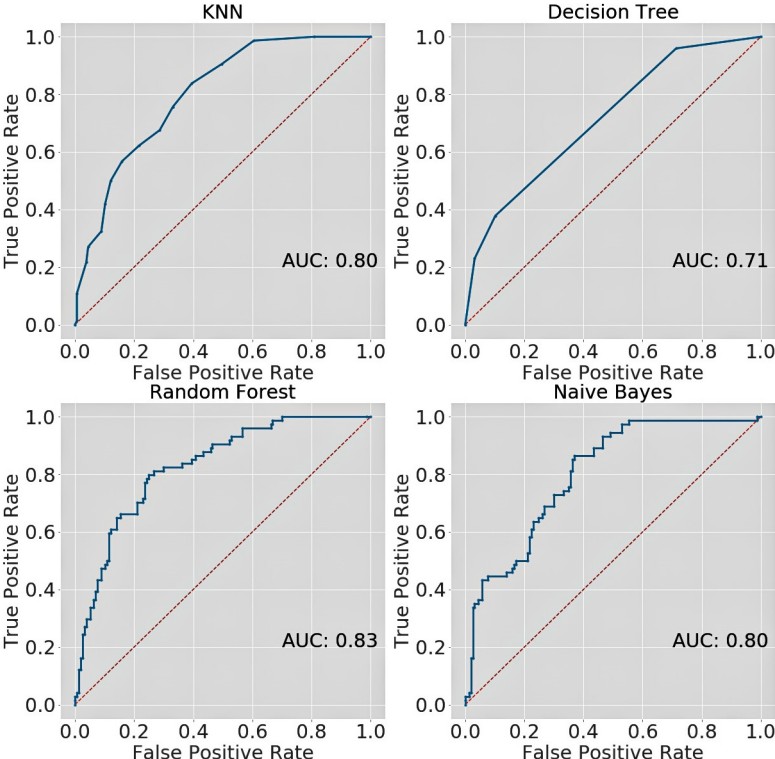

**Figure 9.** ROC curve comparison of classifiers on PIMA test set.

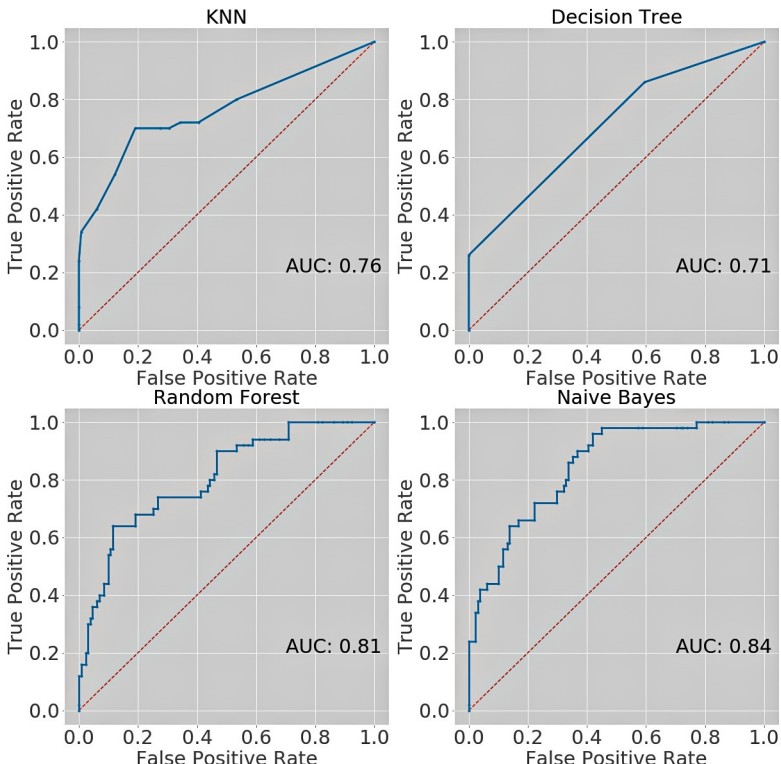

**Figure 10.** ROC curve comparison of classifiers on Kurmitola dataset.

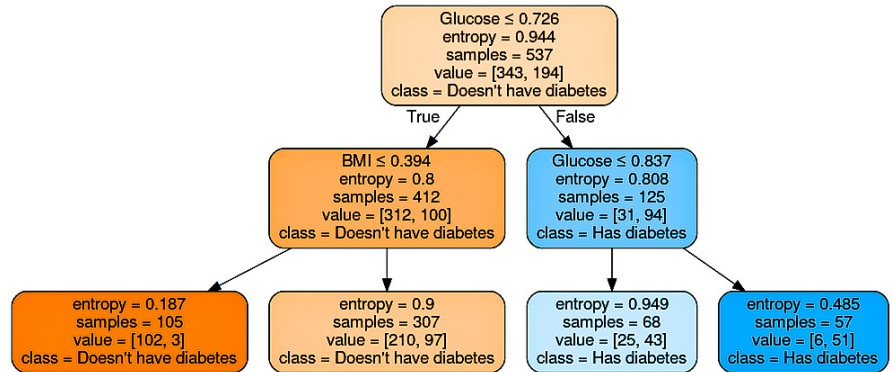

**Figure 11.** Visualization of hyperparameter tuned decision tree with maximum depth = 2.

**Table 7.** Comparison of proposed model with existing works on PIMA dataset.

| SL | Dataset | Authors | Accuracy | Precision | Recall | F1-score | AUC |
|----|---------|---------|----------|-----------|--------|----------|-----|
| 01. | | Islam et al. [30] | 0.78 | 0.89 | 0.80 | — | 0.83 |
| 02. | | Varma et al. [32] | 0.74 | — | — | — | — |
| 03. | PIMA Indian | Gujral et al. [33] | 0.82 | — | — | — | — |
| 04. | | Radja et al. [34] | 0.77 | 0.79 | 0.90 | 0.76 | — |
| **06.** | | **Proposed work** | **0.78** | **0.81** | **0.89** | **0.84** | **0.83** |

As the performances of the classifiers applied to both PIMA Indian and Kurmitola General hospital datasets are depicted in Table 6, classifiers performed well in both cases, although they were only trained on 70% of PIMA dataset. These facts provide a valid proof that diabetes of patients from Bangladesh can be detected using Machine Learning techniques while the training dataset is collected from India.

## 6. Conclusions

Diabetes, although a non-communicable disease, is a serious condition with no cure. It can develop slowly in the body and further enhance the risk of other associated diseases. Obesity, chemical toxins in food, lack of physical exercise, sedentary lifestyle, and poor nutrition are all these risk factors for a person to suffer from diabetes eventually. Taking preventive measures and raising proper awareness can help reduce the risk of this complication. In a developing country like Bangladesh, most of the people have poor knowledge regarding a healthy lifestyle and are unaware of the fact that they are already suffering from diabetes or developing the condition. Early prediction of diabetes will allow a patient to take necessary precautionary measures and control the condition from getting worse.

In order to answer the research question posed in this article, a series of experiments have been conducted. The purpose of these experiments is to evaluate whether female patients in Bangladesh having diabetes can be detected with high confidence using Machine Learning techniques. In Section 5, we have demonstrated that Machine Learning techniques are reliably effective in detecting diabetes. This infers that the unavailability of the dataset will not be a big issue for Bangladesh if Machine Learning techniques can be well applied. We worked on a relatively small dataset from the Bangladesh side.

In future, we plan to collect a more enriched dataset which will help us to predict diabetes detection with higher confidence. In addition to this, we also plan to extend this work to evaluate how complex classifiers based on Artificial Neural Network (ANN) or other deep learning techniques perform when a certain dataset is used for training and another dataset is used for testing. This will help us understand whether such complex classifiers yield better predictions for such purposes. Furthermore, our idea can be generalized to any disease prediction, not just diabetes, especially if there is a data insufficiency problem.

**Author Contributions:** Conceptualization, S.M., B.P. and S.M.M.; data curation, B.P., I.M.S. and E.B.M.; formal analysis and investigation, B.P., S.M.M., S.M. and A.R.; methodology, B.P., S.M.M., E.B.M., I.M.S. and S.M.; writing—original draft preparation, B.P., S.M.M., E.B.M. and I.M.S.; writing—review and editing, S.M.M., E.B.M., S.M. and A.R.; validation, A.R., B.P. and S.M.M.; supervision, S.M. and A.R. All authors read and agreed to the published version of the manuscript.

**Funding:** This research received no external funding.

**Conflicts of Interest:** The authors declare no conflict of interest.

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
