# Peer review of "Evaluating Machine Learning Methods for Predicting Diabetes among Female Patients in Bangladesh"

_information, doi:10.3390/info11080374_

Round 1

Reviewer 1 Report

This manuscript presents an interesting work on a quite interesting topic. Although the classification of diabetes patients based on a known PIMA dataset has been performed and published many times (as also authors present in the related work section), this work has the potential for an innovative research aspect, namely a kind of transfer learning. Authors namely train models on the PIMA dataset but apply them also on their original data collected in Bangladesh. But this aspect of their work is not properly discussed and evaluated based on the experimental results.

I, therefore, recommend authors to formulate more clearly their research question or research focus at the beginning of the paper, focus related work analysis and their results mainly towards this question/goal(s) and finally, properly discuss the experimental results with respect to the research question/goal(s). 

Moreover, I have the following recommendations for further improvements of the manuscript: 

  • Instead "1 in X cases" use "1 out of X cases" (twice on page 2). 
  • The area of Bangladesh is provided with not properly inserted commas.
  • The plural of verbs on some places in the manuscript is missing ("s" at the end of the verb).
  • On page 6 use the "highest correlation" instead of "most correlation".
  • On page 9 use "find suitable values of hyperparameters" instead of "choose hyperparameters".
  • When describing the Naive Bayes algorithm in subsection 4.3.4 formula (6) is unnecessary, instead of it provide the formula for calculation of P(X,Y) based on values of individual parameters P(xi,Y), where you can demonstrate why this algorithm is called Naive. 
  • Subsections 5.1 to 5.4 should be completely rewritten. The current text is redundant because you just repeat what is clear from Table 4. Authors should rather interpret and analyze the results in light of their research question/goal(s). Moreover, they could compare their results with those presented in the related work section. 
  • In Table 4 authors should either remove symbols "%" in the table header or move the decimal point two decimal places to the right in all numbers they provide.
  • I suggest highlighting the best results in Table 4 e.g. by means of bold font. 

Reviewer 2 Report

Decision tree, K-nearest neighbor, Random forest, and Naïve Bayes are used for Predicting Diabetes. Please kindly consider the following comments:

Paper uses several ML methods, why title says one ML method?

+remove citation from abstract, and follow the journal guideline for presenting the abstract.

+state of the art and research gap is presented poorly; elaborate on that.

+state of the art of ML methods in diagnosis is not well presented. 

+why using K-nearest neighbor, Random forest, and Naïve Bayes for this application? what is the strength behind using ML in general and proposed ML methods? studies such as Joloudari et al. Coronary artery disease diagnosis; ranking the significant features using a random trees model." International journal of environmental research and public health 17, no. 3 (2020): 731. uses ML models for diagnosis in comparative studies. such studies must be reviewed in the introduction section to justify the application of ML methods. 

+elaborate on the comparative analysis of ML, presentation, and discussion is weak.

+proofreading is essential, paragraphs are not well connected and it is difficult to follow. Too many paragraphs used. each page should include 4-5 well-connected paragraphs. work on the presentation style.

+conclusions needs improvement to back the results and also include future works. remove citations from conclusions.

+explanin more on the data source and the details of the data

+explain on validations, and how data is divided for testing and training and why?

Round 2

Reviewer 2 Report

Thank you for the revised version.

The results cannot be observed and evaluated due to the low-quality figures. Please improve all the figures. Please make them enlarged and sharper. Please ensure all the figures are reproduced including the methodology figure. Please fix the affiliations and apply native proofreading. Please elaborate more on the details of the data set.
